# Synthesis of Porous Carbon Nitride Nanobelts for Efficient Photocatalytic Reduction of CO_2_

**DOI:** 10.3390/molecules27186054

**Published:** 2022-09-16

**Authors:** Zhiqiang Jiang, Yirui Shen, Yujing You

**Affiliations:** 1School of Materials Science and Chemical Engineering, Ningbo University of Technology, Ningbo 315211, China; 2Zhejiang Institute of Tianjin University, Ningbo 315201, China

**Keywords:** porous materials, carbon nitride, photocatalyst, CO_2_ reduction

## Abstract

Sustainable conversion of CO_2_ to fuels using solar energy is highly attractive for fuel production. This work focuses on the synthesis of porous graphitic carbon nitride nanobelt catalyst (PN-g-C_3_N_4_) and its capability of photocatalytic CO_2_ reduction. The surface area increased from 6.5 m^2^·g^−1^ (graphitic carbon nitride, g-C_3_N_4_) to 32.94 m^2^·g^−1^ (PN-g-C_3_N_4_). C≡N groups and vacant N_2C_ were introduced on the surface. PN-g-C_3_N_4_ possessed higher absorbability of visible light and excellent photocatalytic activity, which was 5.7 and 6.3 times of g-C_3_N_4_ under visible light and simulated sunlight illumination, respectively. The enhanced photocatalytic activity may be owing to the porous nanobelt structure, enhanced absorbability of visible light, and surface vacant N-sites. It is expected that PN-g-C_3_N_4_ would be a promising candidate for CO_2_ photocatalytic conversion.

## 1. Introduction

Significantly increased CO_2_ concentration in the atmosphere has caused problems such as air pollution and global warming in the past decades, posing a serious threat to our future generations [1,2]. In order to alleviate these issues, innovative and sustainable technologies are needed to effectively capture and convert CO_2_. Sustainable conversion of CO_2_ to high value-added products not only helps to reduce the content of CO_2_ in the atmosphere but also promotes the carbon cycle [3,4]. However, due to the lack of efficient, stable, and selective catalysts, the research on CO_2_ photoreduction is still progressing slowly.

Carbon nitride, a low-cost, thermally stable, and nontoxic material with a band gap of 2.7 eV and well-matched requirements of various redox reactions, shows excellent photocatalytic activity [5,6]. However, both low surface area and separation efficiency of photogenerated electron-hole pairs can depress the photoactivity of g-C_3_N_4_ [7,8]. Various methods have been developed to solve these problems, such as nonmetal element doping [9], metal oxide or hydroxides [10,11], ion doping [12,13], noble metal deposition [14,15], controlling morphology [16,17], and loading on carriers [18,19]. Previous research has demonstrated the great potential of g-C3N4 in photocatalytic fields, including carbon dioxide reduction, water splitting, organic pollutant degradation, and organic transformations [5,20].

In bulk g-C_3_N_4_, the stacked 2D single layers are held together in place by weak van der Waals forces of attraction [21]. Aiming to provide more reaction sites, exfoliation has been of particular interest for modifying g-C_3_N_4_ in recent years. g-C_3_N_4_ is treated with thermal exfoliation [20], concentrated acid treatment [22], and ultrasonic exfoliation [23] to synthesize nanoribbon or nanobelt samples. In this study, we used melamine as the precursor and a mixed solution of distilled water and ethylene glycol as the solvent. Melamine molecularly dissolved in the mixed aqueous solution at elevated temperature and polymerized into porous carbon nitride nanobelt (PN-g-C_3_N_4_), as displayed in Figure 1. The thin layer may be formed due to the polyols introduced into the interlayer [24] in the hydrothermal process and ultrasonic-induced exfoliation [25] in the washing stage of supramolecular precursor. Meanwhile, during thermal calcination process the released gas and volume shrinkage of precursor would create many pores on the layers, finally producing porous few-layer C_3_N_4_. The morphologies, microstructures, and physicochemical properties of the photocatalyst were studied. CO_2_ was used as the raw material to evaluate PN-g-C_3_N_4_ photocatalytic performance under visible light and simulated sunlight. The obtained PN-g-C_3_N_4_ was confirmed to be an efficient photocatalyst in the conversion of CO_2_.

## 2. Results

### 2.1. SEM Analysis

Figure 2 presents the morphologies and micro-structures of PN-g-C_3_N_4_ and g-C_3_N_4_ samples. The as-obtained PN-g-C_3_N_4_ nanobelts are characterized with thickness of 30–80 nm with a lateral size of micrometers. It can be seen that PN-g-C_3_N_4_ has loose nanobelt structures with pores in its framework (Figure 2a,b). The existence of a large number of edges and pores in the obtained porous nanobelt structures is extremely important for improving the photochemical and catalytic performance of carbon nitride. On the contrary, g-C_3_N_4_ has bulk structure (Figure 2c,d), which is formed by lamellar structures stacking with each other.

The nitrogen adsorption–desorption isotherms of the samples are presented in Figure 3a. Both g-C_3_N_4_ and PN-g-C_3_N_4_ exhibited a type IV isotherm with a hysteresis loop at P/P_o_ = 0.6–1.0. The pore sizes in PN-g-C_3_N_4_ are about 3–20 nm, which is attributed to the pores formed in the porous nanobelt structures (Figure 3b). The BET surface areas of g-C_3_N_4_ and PN-g-C_3_N_4_ were calculated to be 6.5 m^2^·g^−1^, and 32.94 m^2^·g^−1^, respectively. The specific surface area of PN-g-C_3_N_4_ increases greatly due to its porous nanobelt structures, which is beneficial for the exposure of more active catalytic sites.

### 2.2. IR and UV-Vis DRS Analysis

The FTIR spectrum for g-C_3_N_4_ and PN-g-C_3_N_4_ (Figure 4) showed a peak at 807 cm^−1^ typical for the out-of-plane bending mode of heptazine rings, whilst peaks locked between 800 and 1800 cm^−1^ originated from N-C=N heterorings [26]. The peak at 3000–3500 cm^−1^ corresponded to N-H stretching vibrations. For the PN-g-C_3_N_4_ samples, a new peak centered at 2173 cm^−1^ is found in the spectrum, which is assigned to an asymmetric stretching vibration of C≡N triple bond. The other change was the decrease in the intensity of the N-H stretching peaks between 3000 and 3300 cm^−1^. The results suggest the synthesis of PN-g-C_3_N_4_ decreases the concentration of N-H groups and introduces C≡N groups. The existence of C≡N groups in PN-g-C_3_N_4_ is supposed to increase the electron delocalization and adjust band structures, beneficial for visible-light absorption and photon-generated carrier separation [27].

The optical absorption properties of the photocatalyst have a great effect on the photocatalytic performance. In order to investigate the optical absorption properties of the samples, diffuse reflectance absorption spectra were recorded on UV-Vis system. The optical absorption spectra of g-C_3_N_4_ and PN-g-C_3_N_4_ are displayed in Figure 5a. The absorption edge of g-C_3_N_4_ was at around 460 nm. However, the absorption spectrum of PN-g-C_3_N_4_ extends to the more visible light region from 420 nm to 800 nm. The results demonstrate that PN-g-C_3_N_4_ has enhanced optical adsorption of the visible light, which ascribe to large number of edges and pores in the obtained porous nanobelt structures. The band gaps of g-C_3_N_4_ and PN-g-C_3_N_4_ are presented in Figure 5b. The band gap of PN-g-C_3_N_4_ (2.50 eV) is lower than that of g-C_3_N_4_ (2.68 eV). These changes are related to quantum confinement effect, due to excitation into the lower energy defect states [28]. The relatively low band gap of PN-g-C_3_N_4_ allows it to absorb a good number of photons in the visible domain of the solar spectrum, which is the most important for an effective photocatalyst.

### 2.3. XRD and XPS Analysis

The XRD pattern for pristine g-C_3_N_4_ (Figure 6a) showed two characteristic peaks at 13.0° and 27.4°, which can be assigned to the (100) and (002) crystal planes of g-C_3_N_4_, representing in-plane packing and interfacial stacking of g-C_3_N_4_ sheets, respectively [29]. The peak at 27.4° of PN-g-C_3_N_4_ is weaker and wider, suggesting that the interlayer structure of g-C_3_N_4_ has been weakened, which agrees well with the changes in the micro-morphology.

The survey XPS spectra of g-C_3_N_4_ and PN-g-C_3_N_4_ samples are shown in Figure 6b. The XPS data showed a decrease in the N/C ratio from 1.29 (g-C_3_N_4_) to 1.20 (PN-g-C_3_N_4_) on the surface, suggesting the introduction of surface N defects. High-resolution XPS peaks of C1s spectra of the g-C_3_N_4_ sample in Figure 6c are deconvoluted into three peaks for C-C (285.0 eV), C-N (286.5 eV), and N-C=N (288.5 eV) bonds. Moreover, the high-resolution XPS peaks of N1s spectrum (Figure 6d) is deconvoluted into four peaks. The first peak at 398.9 eV represents the C=N-C bond. The other peaks at 400.3, 401.5, and 404.8 eV belongs to N-(C)_3_, C-N-H, and π excitation bonds, respectively (Figure 6d). Compared to g-C_3_N_4_, PN-g-C_3_N_4_ showed a slight shift in all the peaks of C1s (0.1–0.2 eV) and N1s (0.1–0.3 eV) spectra, which may be caused by the defects in the carbon nitride network. Interestingly, the intensity of C-C peaks of PN-g-C_3_N_4_ slightly increased. The peak area ratios between C-C and N-C-N of C1s spectra were calculated to be 0.18 and 0.25 for g-C_3_N_4_ and PN-g-C_3_N_4_ samples, respectively. Similarly, the peak area ratios between C=N-C and N-(C)_3_ peaks in the N1s spectra were determined to be 4.5 (g-C_3_N_4_) and 3.6 (PN-g-C_3_N_4_), respectively. It is strong evidence that C=N-C vacancies are formed on the surface of PN-g-C_3_N_4_, which can act as entrapping points for charges, yielding longer lifetimes for the charge carrier photoexcitons [30].

### 2.4. Photocatalytic Performance

The photocatalytic activities of as-prepared g-C_3_N_4_ and PN-g-C_3_N_4_ are shown in Figure 7. In the range of 420–800 nm, simulated sunlight has the similar profile with visible light. However, simulated sunlight has energy distribution at UV zone (360–420 nm) and NIR zone (820 nm) while these parts of visible light are cut off. Remarkably, the sample PN-g-C3N4 exhibits an excellent CO evolution rate (29.8 μmol·h^−1^·g^−1^), which is about 5.7 times that of g-C_3_N_4_ (5.2 μmol·h^−1^·g^−1^) under the visible light (Figure 7a). The CO evolution rate under simulated sunlight catalyzed by PN-g-C_3_N_4_ is 52.6 μmol·h^−1^·g^−1^, which is about 6.3 times that of g-C_3_N_4_ (8.3 μmol·h^−1^·g^−1^). The results confirmed porous nanobelt structures of PN-g-C_3_N_4_ can extremely enhance the specific surface area and provide more space for mass transfer and reaction, which in turn improves the photocatalytic activity of the samples. The superior activities of PN-g-C_3_N_4_ can also be attributed to enhanced visible-light absorption and N defects.

As presented in Figure 7b, the yield of CO is stable without any significant deactivation after five cycles (15 h illumination), which indicates high photostability of PN-g-C_3_N_4_ for the CO_2_ reduction. It is worth noting that no other gas products such as CH_3_OH or CH_4_ generated by PN-g-C_3_N_4_ were detected by gas chromatography.

Electrochemical tests were performed in a three-electrode cell with a g-C_3_N_4_-coated working electrode to further understand the dynamics of electron transfer at the PN-g-C_3_N_4_ surface. Figure 8a shows the current of the electrochemical cell with pulsed light excitation. Under visible light illumination, both g-C_3_N_4_ and PN-g-C_3_N_4_ generated significant photocurrent, implying efficient photogeneration of charge carriers in both materials that is then transferred to the working electrode. Furthermore, the PN-g-C_3_N_4_ showed higher photocurrent intensity than that of g-C_3_N_4_, suggesting the higher separation rate of photogenerated charge carriers in the PN-g-C_3_N_4_. Additionally, the photocurrent can reproducibly increase and recover in every on-off cycle of irradiation, demonstrating the high stability in practical applications. The photogenerated electrons and holes are likely separated more efficiently in PN-g-C_3_N_4_ than in g-C_3_N_4_. To test this, photoluminescence (PL) measurements were performed to study the separation of photogenerated electrons and holes in g-C_3_N_4_ and PN-g-C_3_N_4_. Figure 8b displays the PL spectra of the two samples under 380 nm excitation at room temperature. The strong emission peak of g-C_3_N_4_ around 465 nm was derived from the direct band transition. By contrast, the PL intensity of PN-g-C_3_N_4_ was 65% lower, indicating the higher efficiency in separation of the photogenerated charge carriers. Furthermore, the morphology change from multi-layer structure (g-C_3_N_4_) to thin nanobelts (PN-g-C_3_N_4_) would shorten the distance for the photogenerated electrons to reach the surface, thus facilitating the charge separation.

The VB XPS spectra (Figure 8c) shows that the band gap of g-C_3_N_4_ and PN-g-C_3_N_4_ between the valence band (VB) and Fermi level (E_f_) are 2.38 and 2.25 eV [31], respectively. The Mott–Schottky plot (Figure 8d) of g-C_3_N_4_ and PN-g-C_3_N_4_ illustrates that the flat band potentials are −0.80 and −0.69 V, versus the saturated calomel electrode (SCE). The Fermi levels of g-C_3_N_4_ and PN-g-C_3_N_4_ are −0.58 and −0.47 V (vs. NHE) [32]. Therefore, the CB and VB potentials of g-C_3_N_4_ can be calculated to −0.88 and 1.80 eV, respectively, while the CB and VB potentials of PN-g-C_3_N_4_ were equal to −0.72 and 1.78 eV, respectively [33]. The potential position change between g-C_3_N_4_ and PN-g-C_3_N_4_ is shown in Figure 9, and the band gap structures and charge migration of g-C_3_N_4_ and PN-g-C_3_N_4_ are illustrated.

### 2.5. Possible Mechanism

The possible reaction mechanism is discussed for photocatalytic CO2 reduction with water into CO over PN-g-C_3_N_4_ as depicted in Figure 10. Generally, the photocatalytic CO_2_ reduction reaction involves the following three steps: (i) CO_2_ adsorption and activation; (ii) photo-produced charge carriers’ excitation and transfer to the catalyst surface; and (iii) photocatalytic reaction [34]. Upon illumination with light, the photocatalyst generated electrons (e^–^) in the CB and holes (h^+^) in the VB, as shown in Equation (1). Further, the e^–^ are exploited to reduce CO_2_ to its radical (CO_2_^•−^), as shown in Equation (2) [7]. The water (H_2_O) oxidation arises at VB of the catalyst to produce the energetic protons (H^+^) and oxygen, as shown in Equation (3). The CO_2_^•−^, H^+^, and e^–^ further boosted the rate CO generation, as shown in Equation (4). In the present investigation, CO was developed, which involves an 2e^–^/2H^+^ reduction process [35], as shown in Equation (5). In semiconductors, the numerous e^–^ and H^+^ transfer by proton-coupled electron transfer mechanism is feasible for multi e^–^ reduction reaction.
PN-g-C_3_N_4_ + h*υ* → PN-g-C_3_N_4_^*^ + h^+^+ e^−^(1)
CO_2_ + e^−^ + h*υ* → CO_2_^•−^(2)
H_2_O + 2h^+^ + h*υ* → (½)O_2_ + 2H^+^(3)
CO_2_^•−^ + 2H^+^ + e^−^ + h*υ* → CO + H_2_O(4)
CO_2_ + 2H^+^ + 2e^−^ + h*υ* → CO + H_2_O(5)

## 3. Materials and Methods

### 3.1. Materials

The chemical reagents used for the synthesis of PN-g-C_3_N_4_ were commercially available reagents. Melamine, hydrochloric acid, and ethylene glycol were purchased from Sigma-Aldrich. All the chemicals were used as received without further purification.

### 3.2. Synthesis of Catalysts

The PN-g-C_3_N_4_ was synthesized using a simple one-pot hydrothermal method. First, in a typical synthesis procedure, melamine (2 g, 99%) was dissolved in the mixture of distilled water (40 mL) and ethylene glycol (20 mL, 99%) to make a clear solution at 60 °C. Then, 2.4 mL concentrated hydrochloric acid (36.5%) was added into 60 mL of this solution by stirring for 10 min. Then, the mixed solution was transferred into a Teflon-lined autoclave and heated at 150 °C for 12 h. The mixture was filtered to remove the solvent and the precipitate was washed several times with ethanol and deionized water under ultrasonication, followed by drying overnight at 60 °C in vacuum oven. The resulting solid was heated at 600 °C for 2 h with a heating rate of 3 °C·min^−1^. The g-C_3_N_4_ was synthesized by directly heating melamine at 500 °C for 2 h with a heating rate of 3 °C·min^−1^.

### 3.3. Characterization

XRD spectra were recorded on a Bruker D8 Advance diffractometer (Cu Ka radiation). The IR spectra were collected with a Thermo Nicolet iS50 FTIR spectrometer, equipped with an attenuated total reflection (ATR) setup. Diffuse reflectance absorption spectra were recorded on a Varian Cary 4E UV-Vis system equipped with a Labsphere diffuse reflectance accessory. X-ray Photoelectron Spectroscopy (XPS) experiments were performed on Thermo ESCALAB 250 using monochromatized Al Kα at hυ = 1486.6 eV. Bandgap energy (Eg) of the g-C_3_N_4_ and PN-g-C_3_N_4_ samples was calculated according to the formula below:(αhv)^1/n^ = C(hυ − Eg)
where α, υ, and C are the absorption coefficient, light frequency, and a constant, respectively. The parameter n is a pure number corresponding to different electronic transitions (n = 2 or 1/2 for indirect-allowed or direct-allowed transitions, respectively.

### 3.4. Photoactivity Meaasurements

The photocatalytic CO_2_ reduction test was performed using a batch process under visible light with a 300 W Xenon lamp. In addition, a 420 nm cutoff filter was used to prevent the UV light and Am1.5 filter was used to simulate solar spectral. In this experiment, as-prepared photocatalyst (10 mg) was ultrasonically dispersed in 10 mL of deionized water using a 50 mL round-bottom quartz photo-reactor. Then, the reactor was tightly closed with a silicone rubber septum and the solution was saturated with CO_2_ gas for 30 min before the light illumination. After illumination, the gaseous product such as CO was analyzed by gas chromatography.

## 4. Conclusions

In summary, porous g-C_3_N_4_ nanobelts were synthesized via a facile hydrothermal method. The obtained PN-g-C_3_N_4_ had abundant pores and edges, high specific surface areas, and possessed C≡N groups and vacant N on the surface, which dramatically improved the photocatalytic performance. This catalyst displayed enhanced optical absorption in the visible range. It efficiently and selectively catalyzed CO_2_ reduction to CO under both visible light and simulated sunlight illumination. Enhanced visible light absorption and the existence of vacant N-sites on the surface also contributed to the photocatalytic activity of PN-g-C_3_N_4_. The successful synthesis of PN-g-C_3_N_4_ opens up a new way to improve the photochemical performance of carbon nitride-based catalyst.

## Figures and Tables

**Figure 1 molecules-27-06054-f001:**
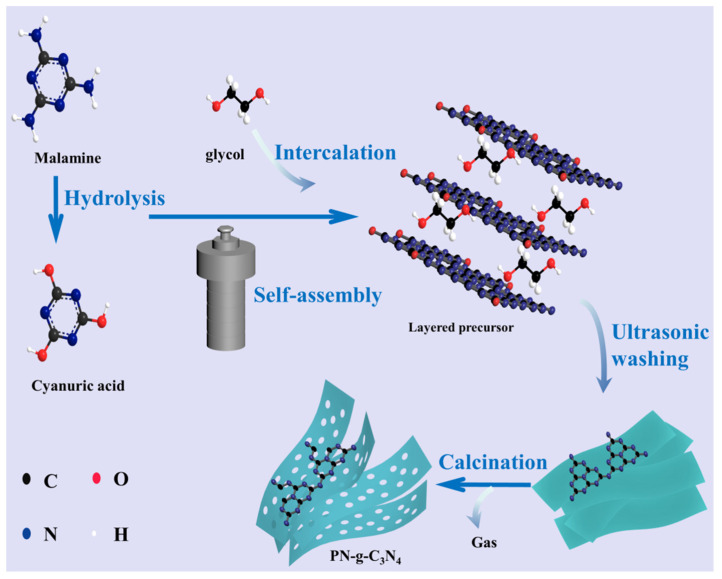
Schematic diagram on the fabrication of PN-g-C_3_N_4_ photocatalyst.

**Figure 2 molecules-27-06054-f002:**
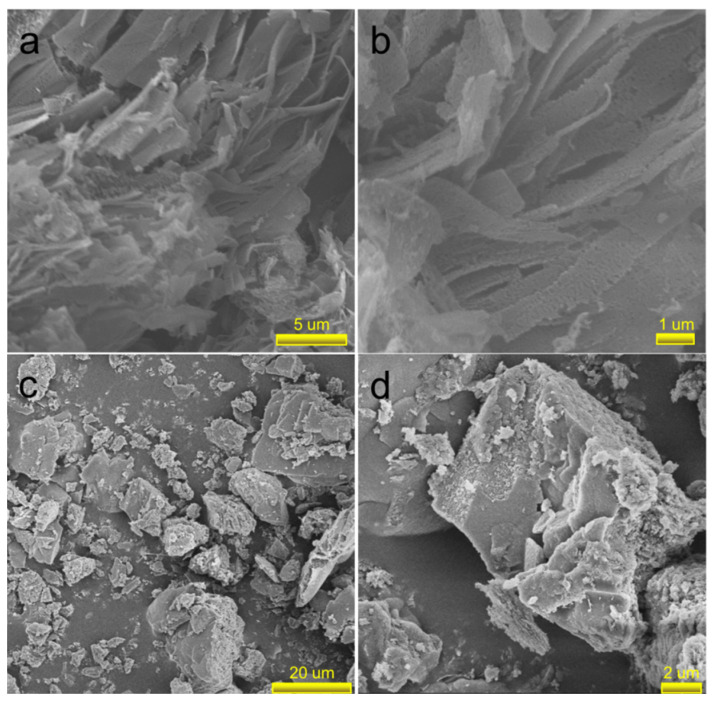
SEM images of PN-g-C_3_N_4_ (**a**,**b**) and g-C_3_N_4_ (**c**,**d**) samples.

**Figure 3 molecules-27-06054-f003:**
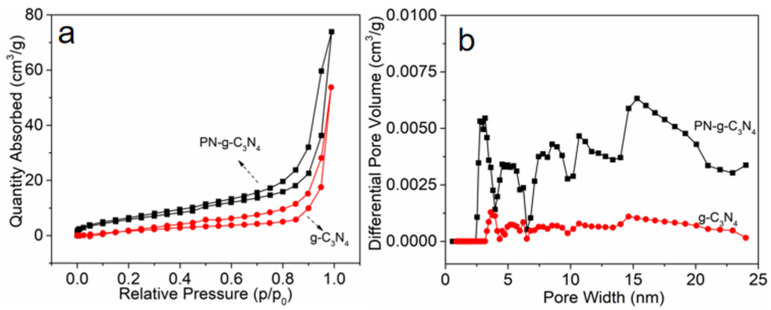
N_2_ adsorption/desorption isotherms (**a**) and BJH pore size distribution of g-C_3_N_4_ and PN-g-C_3_N_4_ (**b**).

**Figure 4 molecules-27-06054-f004:**
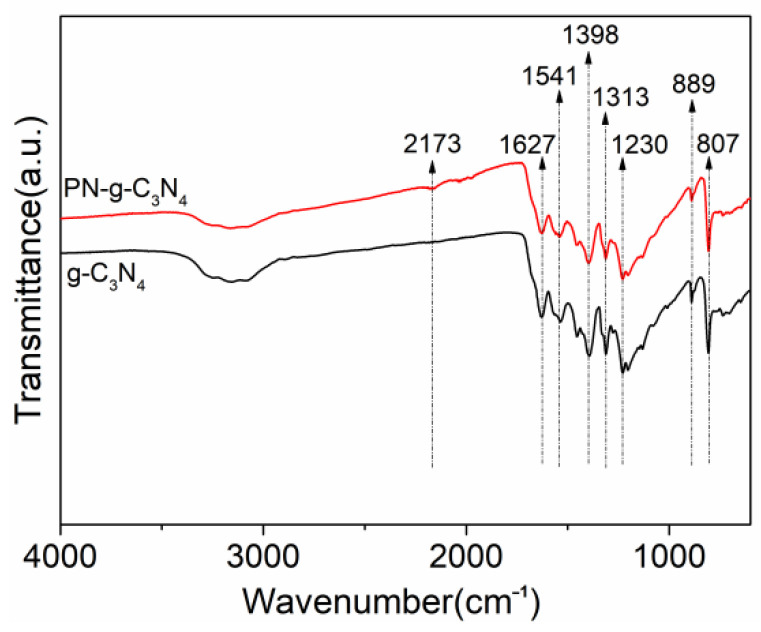
FTIR spectra of g-C_3_N_4_ and PN-g-C_3_N_4_ samples.

**Figure 5 molecules-27-06054-f005:**
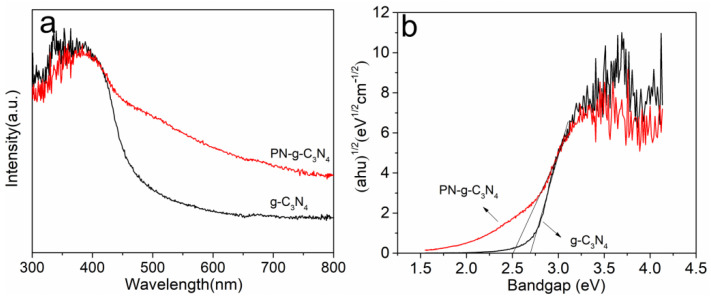
UV-Vis DRS (**a**) and band gaps (**b**) of g-C_3_N_4_ and PN-g-C_3_N_4_ samples.

**Figure 6 molecules-27-06054-f006:**
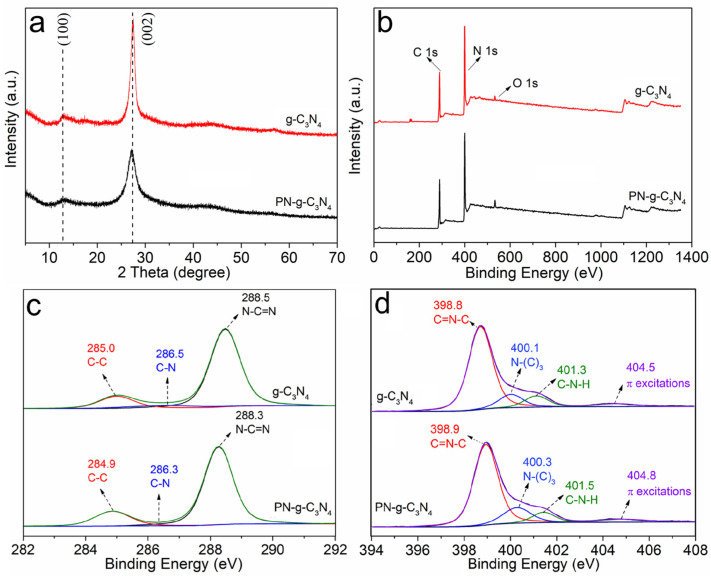
XRD spectra (**a**), XPS survey spectra (**b**), high-resolution C1s (**c**), and N1s (**d**) of g-C_3_N_4_ and PN-g-C_3_N_4_.

**Figure 7 molecules-27-06054-f007:**
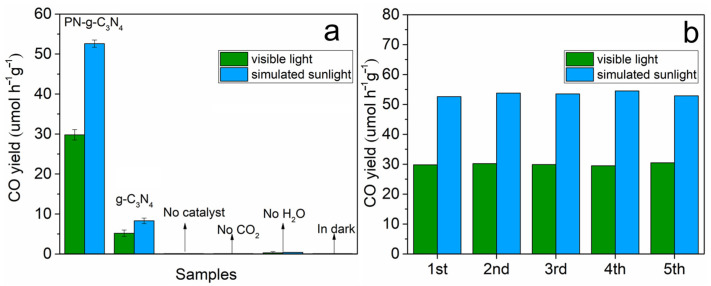
(**a**) Photocatalytic activity of CO_2_ reduction of g-C_3_N_4_ and PN-g-C_3_N_4_ samples under visible light illumination and simulated sunlight with controls. (**b**) Photocatalytic cycle test of PN-g-C_3_N_4_.

**Figure 8 molecules-27-06054-f008:**
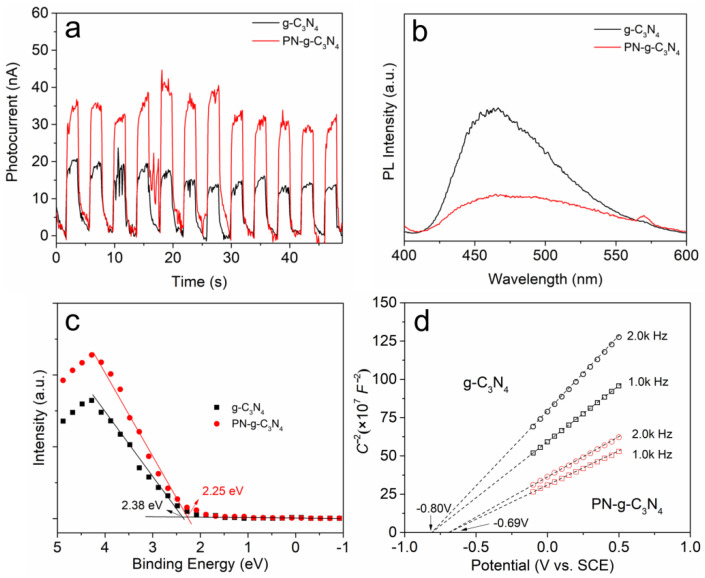
Transient photocurrent response (**a**), PL spectra (**b**), Valence band XPS spectra (**c**), and the Mott–Schottky plot (**d**) of g-C_3_N_4_ and PN-g-C_3_N_4_ samples.

**Figure 9 molecules-27-06054-f009:**
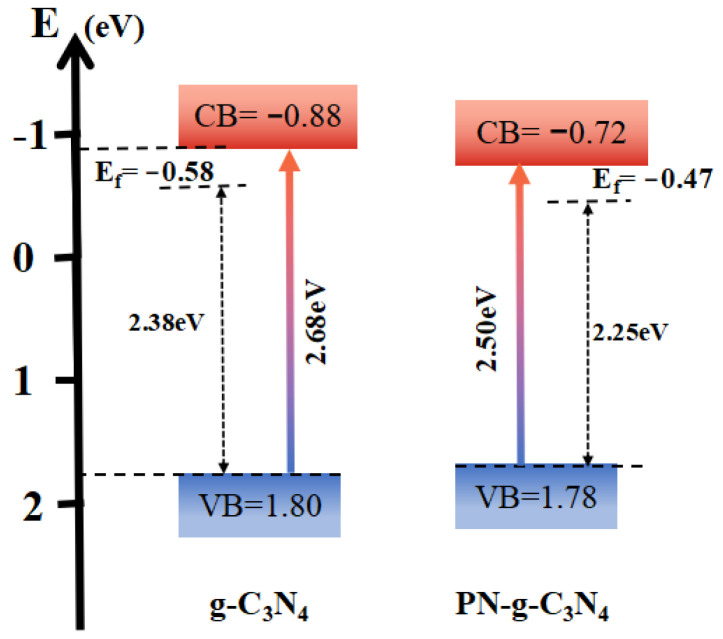
The schematic illustration of the band gap structures of g-C_3_N_4_ and PN-g-C_3_N_4_ samples.

**Figure 10 molecules-27-06054-f010:**
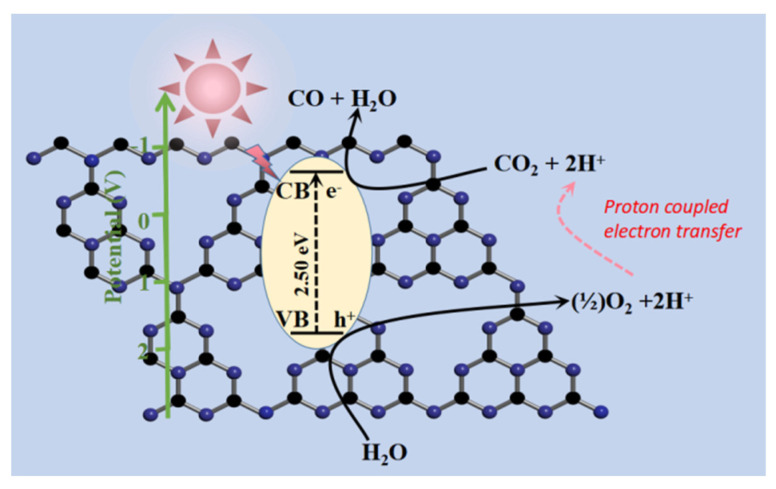
A possible photocatalytic CO_2_ reduction mechanism for CO production.

## Data Availability

Not applicable.

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
