# Peer review of "Synthesis of Porous Carbon Nitride Nanobelts for Efficient Photocatalytic Reduction of CO_2"

_molecules, 2022, doi:10.3390/molecules27186054_

Round 1

Reviewer 1 Report

In this manuscript, the authors synthesized  porous carbon nitride nanobelts for CO2 reduction. However, there are many issues should be cleared before publications. So, I suggest the author to revise the manuscript carefully for resubmission. The detailed comments as follows:

1. The generation mechanism  of nanobelts should be discussed.

2. Their PL spectra should be discussed in manuscript.

3. Photo-charge generation and separation  should be discussed.

4. Detailed mechanism of CO2 reduction should be investigated.

5. 13CO2 should be used to check the CO generation from CO2 reduction.

6. There are many literatures about carbon nanobelts, what is the difference with author's method. 

Reviewer 2 Report

Manuscript ID: molecules-1886014

In the submission titled “Synthesis of porous carbon nitride nanobelts for efficient photocatalytic reduction of CO2”, Jiang et al. reported a study for the development of new-type graphitic carbon nitride with the nano-belt shape and applied the photocatalytic reaction for the CO2 reduction. The authors performed the systematic characterizations of as-synthesized photocatalyst by using the various experimental techniques and demonstrated that employing the photocatalyst in the photoreduction of CO2 showed the excellent catalytic efficiency. This field is certainly emerging and thus this study looks interesting. Overall, I highly recommend the publication of this paper, but the following questions and comments should be addressed by the authors.

(i) In the introduction part, the authors should consider to provide the prospective audience a better background information for the g-C3N4–based photocatalytic reactions. The following literatures could serve this purpose on some aspects; Appl. Catal. B, 2015, 176–177, 44–52 and Catal. Sci. Technol., 2021, 11, 6401–6410.

(ii) Based on the SEM image, the authors explained that “PN-g-C3N4 has loose nanobelt structures with pores in its framework.”. However, it seems that the PN-g-C3N4 has a micrometer size in terms of the long axis. The author should give the clearer explanation for that.

(iii) Figure 7 is one of the key results in this study and clearly displays the catalytic performances with the experimental errors. As I read the experimental section, I can catch that the authors used two different light sources called as visible light and simulated sunlight. For the potential readers, the author should quantitatively provide the spectral window of each light source in terms of wavelength.

Round 2

Reviewer 1 Report

It is confused about the reduction of CO2 reduction described in the manuscript. From the Fig. 7b, the band structure shows that the potential of CB is lower than the reduction potential of CO2/CO (-0.52 V). So, the authors should do more investigations on mechanism of CO2 reduction.
